# Synthesis, Characterization, and Study of the Antimicrobial Potential of Dimeric Peptides Derived from the C-Terminal Region of Lys^49^ Phospholipase A_2_ Homologs

**DOI:** 10.3390/toxins16070308

**Published:** 2024-07-05

**Authors:** Gabriel F. H. Bicho, Letícia O. C. Nunes, Louise Oliveira Fiametti, Marcela N. Argentin, Vitória T. Candido, Ilana L. B. C. Camargo, Eduardo M. Cilli, Norival A. Santos-Filho

**Affiliations:** 1Instituto de Química de Araraquara, Universidade Estadual Paulista (UNESP), Av. Prof. Francisco Degni, 55-Jardim Quitandinha, Araraquara 14800-060, SP, Brazil; gabriel.hispagnol@unesp.br (G.F.H.B.); leticia.catarin@unesp.br (L.O.C.N.); louise.fiametti@unesp.br (L.O.F.); eduardo.cilli@unesp.br (E.M.C.); 2Faculdade de Ciências Farmacêuticas, Universidade Estadual Paulista (UNESP), Rodovia Araraquara Jaú, Km 01-s/n-Campos Ville, Araraquara 14800-903, SP, Brazil; 3Instituto de Física de São Carlos, Universidade de São Paulo (USP), Av. João Dagnone, 1100-Jardim Santa Angelina, São Carlos 13563-120, SP, Brazil; marcela.argentin@usp.br (M.N.A.); vitoriatcandido@usp.br (V.T.C.); ilanacamargo@ifsc.usp.br (I.L.B.C.C.)

**Keywords:** p-BthTX-I, PLA_2_-like, antimicrobial peptides

## Abstract

Currently, the search for new alternatives to conventional antibiotics to combat bacterial resistance is an urgent task, as many microorganisms threaten human health due to increasing bacterial resistance to traditional medicines. Thus, new molecules such as antimicrobial peptides have emerged as promising alternatives because of their low induction of resistance and broad spectrum of action. In this context, in the past few years, our research group has synthesized and characterized a peptide derived from the C-terminal region of the Lys^49^ PLA_2_-like BthTX-I, named p-BthTX-I. After several studies, the peptide (p-BthTX-I)_2_K was proposed as the molecule with the most considerable biotechnological potential. As such, the present work aimed to evaluate whether the modifications made on the peptide (p-BthTX-I)_2_K can be applied to other molecules originating from the C-terminal region of PLA_2_-like Lys^49^ from snake venoms. The peptides were obtained through the solid-phase peptide synthesis technique, and biochemical and functional characterization was carried out using dichroism techniques, mass spectrometry, antimicrobial activity against ESKAPE strains, hemolytic activity, and permeabilization of lipid vesicles. The antimicrobial activity of the peptides was promising, especially for the peptides (p-AppK)_2_K and (p-ACL)_2_K, which demonstrated activity against all strains that were tested, surpassing the model molecule (p-BthTX-I)_2_K in most cases and maintaining low hemolytic activity. The modifications initially proposed for the (p-BthTX-I)_2_K peptide were shown to apply to other peptides derived from Lys^49^ PLA_2_-like from snake venoms, showing promising results for antimicrobial activity. Future assays comparing the activity of the dimers obtained through this strategy with the monomers of these peptides should be carried out.

## 1. Introduction

Microbial resistance to currently used antimicrobial agents is a global public health challenge [1,2]. Therefore, in 2017, the World Health Organization (WHO) published a global list of antibiotic-resistant bacteria that should be given priority in the investigation of new drugs. The group of bacteria that are most common in healthcare infections are known as ESKAPE bacteria (*Enterococcus faecium*, *Staphylococcus aureus*, *Klebsiella pneumoniae*, *Acinetobacter baumannii*, *Pseudomonas aeruginosa*, and *Enterobacter* spp.) and are among the pathogens of the WHO’s list [3]. Therefore, the search for new molecules has become urgent, and antimicrobial peptides (AMPs) have emerged as promising alternatives. AMPs offer a broad spectrum of action, high specificity, and low propensity of selecting microbial resistance [2,4,5].

The Lys^49^ PLA2-like protein, which belongs to the phospholipase A_2_ (PLA_2_) class of toxins, represents a catalytically inactive subgroup due to the presence of a lysine residue at position 49. Despite their catalytic inactivity, Lys^49^ PLA_2_-like proteins exhibit pharmacological activities, such as myotoxic, antifungal, and antibacterial functions, attributed to the sequence in the C-terminal region [6]. Several Lys^49^ PLA_2_ homologs have been described, including myotoxin II (Mt-II) from *Bothrops asper*, bothropstoxin I (BthTX-I) from *Bothrops jararacussu*, K49 phospholipase A_2_ (AppK) from *Agkistrodon piscivorus piscivorus*, and myotoxin from *Agkistrodon contortrix laticinctus* (ACL) [7,8,9,10]. A study led by Santos-Filho et al. (2015) [11] with the peptide p-BthTX-I (sequence KKYRYHLKPFCKK), derived from the C-terminal region of bothropstoxin I, showed that the dimeric form of the molecule was important for antimicrobial activity and was initially formed by a disulfide bond to the cysteine residues at position 11. Subsequent studies evaluated the stability against proteases present in blood plasma, as well as the importance of residues 1 and 2. These studies demonstrated that lysine residues at positions 12 and 13 were quickly degraded, not contributing to the activity of the molecule. Regarding residues 1 and 2, peptides without Lys 1 and 2 showed loss of antimicrobial activity, demonstrating that the N-terminal region is important for assigning charge to the molecule [12].

Considering the importance of dimerization in this class of molecules [11], alternative models that investigated the impact of the position of dimerization were proposed, showing that the homodimer formed through a lysine instead of a cysteine and the deletion of residues 12 and 13 was shown to be equal or more active against certain strains of bacteria than the original molecule [1]. This dimeric form [des-Cys^11^, Lys^12^, Lys^13^-(p-BthTX-I)_2_K] was considered the most promising by the authors [1,12,13] since dimerization can be carried out directly in the resin using Fmoc-Lys(Fmoc)-OH, thus preventing the oxidation of Cys residues, which saves time and shortens the synthesis step, increasing the yield of the crude peptide and stability of the peptide.

Thus, the present work aimed to evaluate whether the modifications proposed by Santos-Filho et al. 2021a [1] can be applied to other synthetic peptides derived from the C-terminal region of PLA_2_ Lys^49^ using the peptide [des-Cys^11^, Lys^12^, Lys^13^-(p-BthTX-I)_2_K] as a model. To do so, modifications were made to peptides previously described in the literature, such as p-ACL, p-MtII, p-AppK, and a modified version of p-MtII, called p-EM-2 [14,15,16,17]. The proposed modifications were expected to enhance the antimicrobial activity of these peptides, offering promising alternatives to conventional antimicrobial agents.

## 2. Results and Discussion

PLA_2_-like proteins from snake venoms have been described in the literature for several decades. Due to the replacement of the Asp residue with the Lys residue at position 49, they do not have catalytic activity. These molecules were initially investigated for their myotoxic potential, attributed to the so-called pharmacological site located in the C-terminal region of this class of proteins [18]. Since then, many PLA_2_-like compounds have been identified and characterized in relation to their myotoxic activity, including myotoxin II (Mt-II), which originated from *Bothrops asper* [7]; bothropstoxin I (BthTX-I), which was found in *Bothrops jararacussu* [8]; the myotoxin ACL, which was found in the venom of *Agkistrodon contortrix lactinctus* [10]; and AppK, which was found in the venom of *Agkistrodon piscivorus piscivorus* [9].

In addition to myotoxic activity, other biological activities of this class of myotoxins, such as antibacterial, antifungal, antiprotozoal, and antitumor activities, have already been reported [19]. Several studies have been carried out with peptides based on the C-terminal region of these proteins, demonstrating that they can replicate the therapeutic potential of the original molecule. Research carried out by Lomonte and collaborators (1994) [14] demonstrated antimicrobial activity with a short peptide of 13 residues located between residues 115 and 129 of *Bothrops asper* myotoxin II, which was named p-MtII. Modifications to optimize this molecule were proposed [20], giving rise to a molecule called p-EM-2 [17], which replaced tyrosine residues with tryptophan and proline with alanine, demonstrating better activity than the original molecule. Another example from the literature is the p-ACL peptide, which has antimicrobial activity, antitumor activity [21], and antileishmanial activity [15]. Antimicrobial activities are also related to the p-AppK peptide [21].

The C-terminal sequence of the PLA_2_-like protein, which originates from snake venom, has great structural similarity and sequence identity, with highly conserved residues, such as cationic and hydrophobic residues. Multiple alignment was performed to analyze this similarity, as shown in Figure 1.

The sequences present highly conserved groups, as Figure 1 shows, such as lysine residues at the N-terminal and C-terminal ends. A study carried out by Santos-Filho et al. (2017) [12] with peptides analogous to p-BthTX-I demonstrated that lysines at positions 12 and 13 are targets of proteases that degrade the C-terminal end and that dimeric peptides synthesized with the deletion of these Lys residues present antimicrobial activity like that of the original molecule and greater stability. In another study, Santos-Filho and collaborators (2021a) [1] also evaluated the importance of lysine residues at positions 1 and 2, and peptides synthesized without these residues exhibited a significant decrease in antimicrobial activity, indicating that the positive charge in the N-terminal region is essential for maintaining antimicrobial activity. It is also possible to speculate about the importance of positive residues at positions 4 and 8 and aromatic residues at position 3, characteristics maintained by all sequences represented in Figure 1.

Another highly conserved residue in the sequences was cysteine at position 11. In a study carried out by Santos-Filho et al. (2015) [11], dimerization of the peptide (p-BthTX-I) through the cysteine residue to form the dimeric peptide (p-BthTX-I)_2_ was found to be crucial for the activity of the molecule, with the dimerization considered responsible for this activity. Furthermore, Bitencourt et al. (2023) [23] observed that the dimerization or dendrimerization of (p-BthTX-I)_2_ was essential for antibacterial activity, as the monomeric structure leads to a total loss of, or significant reduction in, this activity. This characteristic may explain the conservation of this residue in the C-terminal region of the PLA_2_-like protein from snake venom throughout the evolutionary process.

To study the importance of the dimerization position for the antimicrobial activity of the molecule, which initially involved the oxidation of cysteine residues, forming disulfide bonds for the molecule (p-BthTX-I)_2_, another study proposed dimerization through alternative forms. Santos-Filho et al. 2021a [1] dimerized the N-terminal region through a glutamic acid residue, subsequently leading to the oxidation of the cysteine at position 13 of (p-BthTX-I)_2_, forming the cyclic peptide E(p-BthTX-I)_2_, which had decreased antimicrobial activity, demonstrating the importance of the free N-terminal region for charge contribution to the molecule.

Santos-Filho et al. (2021a) [1] also proposed an alternative strategy to dimerization, which consists of replacing cysteine with lysine as a branching bridge for the formation of a homodimer and deleting residues 12 and 13, which, due to the action of proteases, are quickly degraded and become irrelevant to peptide activity. Such modifications have demonstrated, in certain cases, better performance against some strains of microorganisms than the original molecule. They are also easier to obtain because they require fewer steps in peptide synthesis.

The structural similarities of these sequences, as well as the characteristics they share, raise the possibility of modifications previously proposed by Santos-Filho et al. (2021a) [1] for the p-BthTX-I peptide to influence the optimization of the other target peptides in this study.

In this study, the following peptides were synthesized: (1) des-Cys^11^, Lys^12^, Lys^13^-(p-MtII)_2_K, derived from the p-MtII peptide [14]); (2) des-Cys^11^, Lys^12^, Lys^13^-(p-EM-2)_2_K, derived from the pEM-II peptide [17]; (3) des-Cys^11^, Lys^12^, Lys^13^-(p-ACL)_2_K, derived from the p-ACL peptide [15]; and (4) des-Cys^11^, Lys^12^, Lys^13^-(p-AppK)_2_K, derived from the p-AppK peptide [16]. For better understanding and to facilitate discussion of the results, the names of the peptides are abbreviated as (p-MtII)_2_K (1), (p-EM-2)_2_K (2), (p-ACL)_2_K (3), and (p-AppK)_2_K (4). The proposed sequence for the molecules, as well as the names and corresponding analogs, are represented in Figure 2.

### 2.1. Antibacterial Activity

Antimicrobial resistance is a chronic public health problem and occurs when microorganisms acquire resistance to antimicrobial drugs conventionally used for treatment. It is estimated that if no action is taken, approximately 10 million deaths worldwide will occur by 2050, making infectious diseases the leading cause of death worldwide [24].

Bacterial resistance mechanisms to antimicrobial agents are varied and complex, representing a significant challenge in treating infections [3,24]. A common bacterial strategy is to reduce drug accumulation by either reducing the permeability of the outer membrane or increasing the active efflux of drugs through the cell surface, preventing entry or favoring the expulsion of antimicrobial agents [25]. Furthermore, some bacteria produce enzymes capable of inactivating or modifying medications, rendering them ineffective even before they exert their bactericidal effect [26].

Another mechanism is the alteration of target or binding sites, such as the alteration of penicillin-binding or ribosomal proteins, reducing the affinity of antimicrobial agents through these structures. Furthermore, bacteria may develop alternative metabolic pathways that circumvent the effects of antimicrobial agents, such as the absorption of folic acid from the environment, to resist the impact of trimethoprim-sulfamethoxazole. This diversity of mechanisms highlights the importance of comprehensive and multifaceted strategies to combat bacterial resistance, aiming to preserve the effectiveness of antimicrobial agents and ensure successful treatment of infections [25].

The modifications proposed by Santos-Filho et al. (2021a) [1] resulted in a molecule called (p-BthTX-I)_2_K that is more active than the original one (p-BthTX-I). After applying the same dimerization methodology to the molecules (p-MtII)_2_K, (p-EM-2)_2_K, (p-ACL)_2_K, and (p-AppK)_2_K, antimicrobial activity against several multidrug-resistant strains was performed to compare the performance of the dimers. The results of this activity are shown in Table 1. Appendix A show the susceptibility of the strains used in this study to currently approved antibiotics.

In a recent study, Santos-Filho et al. (2021a) [1] tested the dimer (p-BthTX-I)_2_K against the same bacterial strains, except for *S. aureus* ATCC 8095. For clarity and correlation, the previously published [1] results are also presented in Table 1.

In comparison, (p-MtII)_2_K and (p-EM-2)_2_K demonstrated activity against most of the bacterial strains tested, except for *P. aeruginosa* ATCC 27853 (Table 1), with greater efficacy in inhibiting bacterial growth against *S. epidermidis* ATCC 35984, with an MIC of 16 μg.mL^−1^ for both peptides [same as peptide (p-BthTX-I)_2_K] and an MBC of 32 μg.mL^−1^ for (p-MtII)_2_K and 16 μg.mL^−1^ for (p-EM-2)_2_K (both better than the original peptide). The peptide (p-MtII)_2_K was more active than (p-BthTX-I)_2_K against *E. faecium* ATCC 700221 strains, with both lower MICs and lower MBCs; however, (p-BthTX-I)_2_K was superior to the other strains tested. For (p-EM-2)_2_K, the observed activity was greater for all strains except for *K. pneumoniae* ATCC 700603, *A. bauamannii* ATCC 19606, and *P. aeruginosa* ATCC 27853, which achieved equal performance.

The monomer p-EM-2 was previously tested by Santamaria et al. (2005) [17] against the strains *S. aureus* ATCC 23923, demonstrating inhibition of more than 75% of CFUs (colony forming units) at concentrations of 5 and 10 μg.mL^−1^ and approximately 100% of CFUs at a concentration of 50 μg.mL^−1^. For *S. typhimurium* ATCC 14028, the concentration of 5 μg.mL^−1^ had inhibitory effects lower than 75% and approximately 100% CFU inhibition at 50 μg.mL^−1^. The tests carried out by Santamaria et al. (2005) [17] do not allow a direct comparison of performance with that of (p-EM-2)_2_K due to the difference between the types of methodologies and strains used; however, in the present work, the dimeric peptide (p-EM-2)_2_K showed MIC values of 16–32 μg.mL^−1^ in most strains tested, smaller than the monomer MIC values. In general, the antimicrobial activity of (p-EM-2)_2_K was superior to that of (p-MtII)_2_K, demonstrating that the substitution of tyrosine for tryptophan made the molecule more active, as observed by Santamaria et al. (2005) [17] for the peptides p-MtII and p-EM-2. Tryptophan increases the interaction with the membrane and the biological activity of the peptide [27].

The peptides (p-ACL)_2_K and (p-AppK)_2_K showed higher activity than the model molecule (p-BthTX-I)_2_K and the other peptides for all strains tested, both Gram-positive and Gram-negative. It is important to highlight that both peptides showed antibacterial activity against *P. aeruginosa* ATCC 27853, to which activity was not observed by (p-MtII)_2_K and (p-EM-2)_2_K and even to previously reported (p-BthTX-I)_2_K. Furthermore, (p-ACL)_2_K presented better MIC or MBC than (p-AppK)_2_K against most strains, except for *S. epidermidis* ATCC 35984 and *K. pneumoniae* ATCC 700603, which presented the same MIC but a higher MBC, suggesting that the presence of leucine at position 10 may have a more promising influence on antibacterial activity compared to phenylalanine at this position, as seen by Almeida and co-workers (2022) [21]. The superior activity of (p-ACL)_2_K and (p-AppK)_2_K compared to the model peptide (p-BthTX-I)_2_K opens a range of opportunities for new tests with multidrug-resistant bacterial strains.

### 2.2. Circular Dichroism (CD)

To elucidate the secondary structures of the peptides, we examined their distinct features via circular dichroism spectroscopy. As outlined, the far-UV CD image of a disordered structure (random coil) exhibits a positive band at 212 nm and a negative band at 195 nm. Moreover, the far-UV CD spectrum of the β-sheets revealed a negative band at 218 nm and a positive band at 196 nm. For the α-helix, positivity at 195 nm, negativity near 208 nm, and negativity at 222 nm were observed [28].

Many AMPs do not exhibit a clearly defined structure when in aqueous solution; however, they develop secondary conformations, such as alpha-helices and beta-sheets, when interacting with membranes, while others retain a structure without a pattern (random coil). Although CD spectra provide an average of the entire molecule, the technique is not precise enough to identify which specific residues are involved in a given portion [29].

In this work, CD spectra were obtained in phosphate-buffered saline solution (PBS), 60% trifluoroethanol (TFE) solution, and 5 mM lysophosphatidylcholine (LPC) solution. The peptides (p-MtII)_2_K (Figure 3a), (p-EM-2)_2_K (Figure 3b), (p-ACL)_2_K (Figure 3c), and (p-AppK)_2_K (Figure 3d) adopted a disorganized conformation in aqueous solution, which is common in AMPs. In a solution containing the membrane environment LPC, all peptides underwent conformational changes, with a mixture of secondary structures, suggesting some type of interaction with the membrane. However, (p-EM-2)_2_K and (p-AppK)_2_K were apparently structured with helical content. When in a solution containing 60% TFE, which acts as a structuring agent for peptide chains, (p-MtII)_2_K presented a small content of helical structure, while (p-EM-2)_2_K, (p-AppK)_2_K, and (p-ACL)_2_K assumed a conformation of helical structures, with a positive band at 190 nm and two negative bands at 208 nm and 222 nm, respectively.

The molecule described by Santos-Filho et al. (2021a) [1], (p-BthTX-I)_2_K, did not present a defined form either in an aqueous solution, LPC, or TFE, unlike the dimers in this study. The conformational changes in these biomolecules, especially in solutions containing membrane mimetics (LPCs), may suggest a possible mechanism of action in biological membranes [29].

### 2.3. Vesicle Permeabilization

Different models, such as the “barrel-stave” and “toroidal pore” models, have been proposed to describe the interaction of AMPs with lipid membranes (Figure 4).

In the “barrel-stave” model (Figure 4a), the formation of pores occurs with the penetration of peptides into the lipid bilayer, while in the “toroidal pore” model (Figure 4b), the peptides remain associated with the head groups of the phospholipids, inducing a curvature in the membrane. An alternative mode of action is transient pores, which have less organized internal structures [5]. Another mechanism of action is the “carpet-like” (Figure 4c) and “detergent-like” (Figure 4d) model, which suggests membrane permeabilization by the detergent action of peptides. This mechanism involves the aggregation of peptides on the surface of the membrane, like a carpet. When it reaches a concentration threshold, it leads to destabilization of the phospholipid bonds, whereby the membrane is dissolved in a manner-like dispersion, and membrane rupture does not involve a channel formation process [5,6,7,8,9,10,11,12,13,14,15,16,17,18,19,20,21,22,23,24,25,26,27,28,29,30,31,32]. This mechanism involves the interaction of peptides with lipid head groups and does not necessarily interact with the hydrophobic core of the membrane [32]. Despite decades of study, the mechanism of action of many AMPs remains enigmatic. Understanding these mechanisms is crucial for guiding the design of new molecules with antimicrobial activity.

Peptide selectivity is determined by variations in membrane structure and composition. Typically, negatively charged phospholipids are employed to simulate the bacterial membrane, while uncharged (or zwitterionic) phospholipids are used to model the mammalian cell membrane [31,33].

Carboxyfluorescein release or vesicle permeabilization assays are used to investigate the mechanism of action of peptides. Using unilamellar vesicles, it is possible to analyze the interactions between the vesicles and the peptides, obtaining important information about the structure/function relationships of the peptides in membranes [23].

To investigate the interaction of peptides in membranes, two different compositions of lipids were used: (1) vesicles made of 15 mM of 1-palmitoyl-2-oleoyl-sn-glycero-3-phosphocholine (POPC) to resemble the lipid composition of mammalian eukaryotic cells; (2) 12 mM POPC + 3 mM 1-palmitoyl-2-oleoyl-sn-glycero-3-phospho-(10-rac-glycerol) (POPG) to approximate the lipid composition of bacterial membranes, with the presence of a negative charge.

The dimeric peptides (p-MtII)_2_K (Figure 5a), (p-ACL)_2_K (Figure 5c), and (p-AppK)_2_K (Figure 5d) exhibited weak CF release in POPC vesicles, with intensities less than or equal to 10% at concentrations of 4, 32, and 128 μg.mL^−1^, which may suggest weak interactions with eukaryotic cell membranes. This result is in accordance with the result obtained in the hemolytic activity assay (Figure 7), where low toxicity in red blood cells was demonstrated for these molecules.

The leakage of carboxyfluorescein (CF) in (p-EM-2)_2_K was high (Figure 5b), reaching an intensity of 70% at a concentration of 4 μg.mL^−1^ and close to 90% at concentrations of 128 and 32 μg.mL^−1^. This analysis is also in accordance with the hemolytic activity (Figure 7), where the peptide (p-EM-2)_2_K demonstrated the highest capacity to lyse erythrocytes. Most likely, the high degree of Trp residues and greater hydrophobicity conferred a greater affinity for biological membranes, which led to cytotoxicity. The gradual increase observed at concentrations of 4 and 32 μg.mL^−1^ is characteristic of pore formation and may be a strong indication of the mechanism of action of this peptide, as observed in aurein 2.3 [34]. As the concentration increased, an abrupt increase and rapid stabilization were observed, as was the case at a concentration of 128 μg.mL^−1^, indicating that the peptide may have formed a large number of pores, quickly reaching the peak release.

As previously observed with vesicles composed of POPC, the (p-EM-2)_2_K molecule (Figure 6b) also strongly interacts with the bacterial membrane mimetic model composed of 80% POPC + 20% POPG. However, in this case, it is possible to observe greater concentration dependence. The high release of CF in both mammalian and bacterial eukaryotic membrane mimetics suggests low selectivity for the peptide, which is possibly active against biological membranes in general.

The release of CF from bacterial membrane mimetics demonstrated leakage of approximately 10 to 20% for the peptides (p-MtII)_2_K (Figure 6a), (p-ACL)_2_K (Figure 6c), and (p-AppK)_2_K (Figure 6d), with rapid stabilization after reaching the release peak. In addition, they showed a low concentration dependence. The interaction profile combined with the structural conformations adopted by the molecules in membrane mimetics (LPCs) and the fact that these molecules present antimicrobial activity, even if they do not rupture or form permanent pores in the vesicles, may suggest the possible formation of transient pores. However, it is important to highlight that the formation of transient pores only allows the peptides to enter the cell. Further studies must then be carried out to confirm the hypothesis of transient pores, elucidate the mechanism of action itself, and identify the possible intracellular targets of these peptides. Several AMPs kill bacteria using a mechanism independent of membrane destabilizing, and it is known that there are several possible targets on bacterial cells.

### 2.4. Hemolytic Activity

The development of new antimicrobial agents requires, among other things, that the molecule has selective activity against microorganisms and is not toxic to eukaryotic cells. Therefore, to evaluate the toxic potential of the peptides, an in vitro assay was carried out using red blood cells.

According to the hemolytic activity assay (Figure 7), the percentage of lysed red blood cells was less than 10% for the molecules (p-MtII)_2_K, (p-ACL)_2_K, and (p-AppK)_2_K, even at a concentration of 512 μg.mL^−1^, indicating low hemolytic activity. In contrast, the (p-EM-2)_2_K molecule shows high activity, with a concentration capable of lysing 50% of the red blood cells (HC_50_) of approximately 45 μg.mL^−1^.

The modifications proposed by Santamaria et al. (2005) [17] for the monomeric molecule based on the C-terminal region of myotoxin II, p-MtII (KKYRYYLKPLCKK), resulted in the sequence (KKWRWLKALAKK), called p-EM-2. Increased antimicrobial activity was observed, correlated with the number of tryptophan residues added. However, tyrosine substitutions for tryptophan residues also decrease the selectivity of the resulting products for binding to prokaryotic membranes [17]. Even after applying the method proposed by Santos-Filho et al. (2021a) [1] for forming (p-EM-2)_2_K, this dimer showed high toxicity.

A study by Almeida et al. (2022) [21] revealed that the p-ACL and p-AppK peptides are not toxic to red blood cells, with hemolysis percentages less than 7%. Similar results were obtained for the (p-ACL)_2_K and (p-AppK)_2_K dimers, revealing that dimerization did not affect the hemolytic activity of these molecules.

The peptide (p-BthTX-I)_2_K also did not present hemolytic activity against human red blood cells, demonstrating that it is selective against prokaryotic cells (Santos-Filho et al. 2021b) [13], as are (p-MtII)_2_K, (p-ACL)_2_K, and (p-AppK)_2_K.

The structural conformation adopted by (p-EM-2)_2_K of the coiled-coil type in contact with LPC strongly indicates interaction with biological membranes. This was confirmed by the vesicle permeabilization assay, in which a profile characteristic of pore-forming peptides, like the carpet-like mechanism, can be observed at high concentrations. This causes membrane dispersion, leading to rupture, which could explain the hemolytic activity of the molecule.

## 3. Conclusions

The peptides (p-MtII)_2_K, (p-EM-2)_2_K, (p-ACL)_2_K, and (p-AppK)_2_K showed promising results against Gram-positive and Gram-negative bacteria, proving that the modifications proposed by Santos-Filho et al. (2021) [1] can be applied to other peptides derived from the C-terminal region of PLA_2_-like proteins from snake venom. Two molecules with greater potential than the model molecule (p-BthTX-I)_2_K were discovered, namely, (p-ACL)_2_K and (p-AppK)_2_K, which have high antimicrobial activity and low hemolytic activity. Given the promising results, future studies can be proposed to evaluate the possible activity against other microorganisms, both in multidrug-resistant bacterial strains and against yeasts, protozoa, or other microorganisms. Future studies should also be carried out to analyze whether these dimeric peptides are more active than their monomeric units and compare their stabilities in blood serum to fully assess the viability of the best synthesis strategy for this class of molecules.

## 4. Materials and Methods

### 4.1. Peptide Synthesis

Peptides were obtained by solid-phase peptide synthesis (SPPS) [35] using the fluorenylmethyloxycarboxyl (Fmoc) chemistry protocol. For the synthesis of all peptides, Rink amide resin was used with a resin substitution of 0.678 mmol·g^−1^ on a 0.2 mM scale and 2 times excess to anchor the first amino acid Fmoc-Lys(Fmoc)-OH and 4 times excess for coupling from the second amino acid onward.

### 4.2. Purification and Characterization of the Peptides

The synthesized peptides were purified by high-performance liquid chromatography (HPLC) on a Shimadzu chromatograph in semipreparative mode on a C18 reversed-phase column (Phenomenex 2.1 × 25 cm), according to Appendix A. The degree of purity of the fractions was determined on a C18 reversed-phase Shimadzu chromatograph on an analytical column (0.46 × 25 cm) of (Kromassil) in a gradient of 5–95% of solution B in 30 min, with a flow of 1 mL·min^−1^. The profiles of the pure peptides are shown in Appendix A. The solutions used were 0.045% TFA in ultrapure water (solution A) and 0.036% TFA in acetonitrile (solution B).

### 4.3. Mass Spectrometry

Mass spectra were obtained on an LCQ FLEET ThermoScientific, Waltham, MA, USA, mass spectrometer with direct injection in positive detection mode to analyze the correct synthesis of the peptides. Mass spectra of the pure peptides are shown in Appendix A.

### 4.4. Antimicrobial Activity

The compounds were tested against 8 bacterial species: *Staphylococcus epidermidis* (strain ATCC 35984), *S. aureus* (strains ATCC 25923 and ATCC 8095), *Enterococcus faecalis* (strain ATCC 29212), *E. faecium* (strain ATCC 700221), *K. pneumoniae* (strain ATCC 700603), *Escherichia coli* (strain ATCC 25922), *A. baumannii* (strain ATCC 19606)*,* and *P. aeruginosa* (strain ATCC 27853). To carry out the tests, each peptide sample was diluted in DMSO (1%), and a 100× concentrated stock solution was prepared and subsequently diluted 1:100 in Mueller Hinton Cation Adjusted (MHCA) broth (BD, East Rutherford, NJ, USA) according to CLSI (2013) [36]. The peptides were tested at concentrations ranging from 512 to 0.06 μg.mL^−1^. The amount of bacterial inoculum was 5 × 10^5^ CFU/mL, according to the CLSI recommendations. After the end of the incubation time, a visual reading was carried out, which revealed the minimum inhibitory concentration (MIC).

To observe the normal growth of the bacteria, in the positive control, MHCA broth in 1% DMSO was used, with bacteria without the peptide added. In the negative control, only the MHCA broth in 1% DMSO, without bacteria, was added, showing no medium contamination. All tests were performed in duplicate.

After visual reading of the MIC, 100 μL of the contents of the wells corresponding to the MIC, 2× MIC, and 1/2 MIC were inoculated onto an MHCA agar plate using the microdrop technique without streaking. The plate was placed in an incubator for 24 h at 37 °C, followed by a visual reading to determine the minimal bactericidal concentration (MBC), the concentration without colonies.

The susceptibility profile of the strains tested in this study against antibiotics currently in use in hospitals was determined using Phoenix M50 (BD, East Rutherford, NJ, USA) according to the manufacturer’s recommendations using the panels NMIC/ID-470 BD and PMIC/ID-601 for Gram-negative and Gram-positive bacteria, respectively. Results were interpreted following EUCAST recommendations.

### 4.5. Circular Dichroism

To evaluate the secondary structure of the peptides, circular dichroism (CD) spectroscopy was used on a Jasco J-715 spectropolarimeter. CD spectra were obtained in phosphate-buffered saline solution (PBS), 60% trifluoroethanol (TFE) solution, and 5 mM lysophosphatidylcholine (LPC) solution.

### 4.6. Hemolytic Activity

The hemolytic activity of each peptide was determined as described by Santos-Filho et al. (2015) [11] with modifications. The erythrocytes used were washed 3 times with phosphate-buffered saline (PBS) at pH 7.4 at 500× *g* for 5 min and subsequently diluted to a ratio of 1/25 in PBS. The peptides were tested at concentrations between 512 and 1 μg·mL^−1^. After 1 h of incubation at 37 °C, the samples were centrifuged at 500× *g* for 5 min. Aliquots of 80 μL of supernatant were transferred to a microplate, and readings were performed in the emission wavelength range of 540 nm. To determine 100% hemolysis, Triton X-100 at 1% (*v*/*v*) was used. The % hemolysis was calculated using the following equation:$$\%\;hemolysis=\frac{\left(A\;sample-A\;PBS\right)}{\left(A\;Triton-A\;PBS\right)}\;\times \;100$$

The hemolytic concentration capable of lysing 50% of the red blood cells (HC50) was calculated.

### 4.7. Release of Carboxyfluorescein

The permeabilization of lipid vesicles containing carboxyfluorescein was performed using a Jasco FP-8250 spectrofluorimeter. The lipid compositions used were 15 mM 1-palmitoyl-2-oleoyl-sn-glycero-3-phosphocholine (POPC) and 12 mM POPC (80%) and 3 mM (20%) 1-palmitoyl-2-oleoyl-sn-glycero-3-phospho-(10-rac-glycerol) (POPG).

The fluorescence intensity was quantified after adding the peptide (128 μg·mL^−1^, 32 μg·mL^−1^, or 4 μg·mL^−1^) over 195 s, and a 1% Triton X-100 solution was added after 475 s to rupture 100% of the vesicles.

The carboxyfluorescein release graphs were plotted using the equation below:(1)$$\%\;release=\frac{\left(M\;Triton-M\;Sample\right)}{\left(M\;Triton-M\;vesicles\right)}\;\times \;100$$
where M Triton is the average fluorescence value obtained after the addition of Triton, M vesicles is the average of the initial values of the run before the addition of the peptide, and M sample are the fluorescence values obtained at each point after adding the sample.

## Figures and Tables

**Figure 1 toxins-16-00308-f001:**
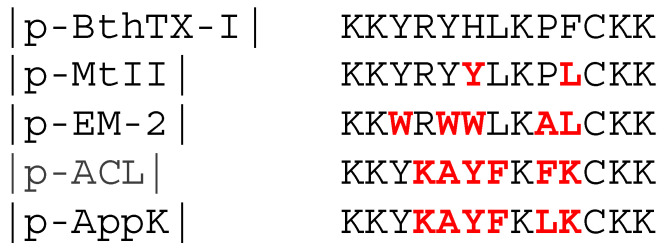
Multiple alignments of peptides derived from amino acids corresponding to residues 115–129 (numeration according to Renetseder et al. (1985 [22]). All sequences were obtained from the NCBI database. Different residues, compared to the BthTX-I sequence, are highlighted in red.

**Figure 2 toxins-16-00308-f002:**
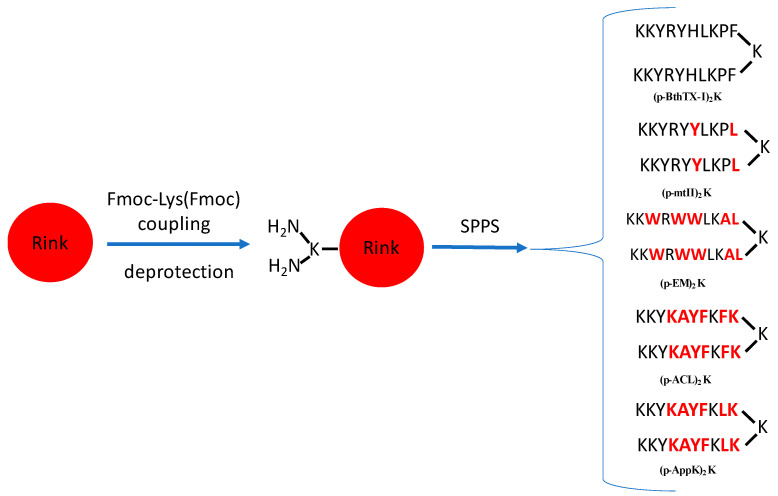
Peptide synthesis strategies based on the molecules des-Cys^11^, Lys^12^, Lys^13^-(p-BthTX-I)_2_K [(p-BthTX-I)_2_K] (Santos-Filho et al., 2021a [1]). The peptides that were synthesized were des-Cys^11^, Lys^12^, Lys^13^-(p-MtII)_2_K [(p-MtII)_2_K]; des-Cys^11^, Lys^12^, Lys^13^-(p-EM-2)_2_K [(p-EM-2)_2_K]; des-Cys^11^, Lys^12^, Lys^13^-(p-ACL)_2_K [(p-ACL)_2_K]; and des-Cys^11^, Lys^12^, Lys^13^-(p-AppK)_2_K [(p-AppK)_2_K], respectively. Different residues comparing to the BthTX-I sequence are highlighted in red.

**Figure 3 toxins-16-00308-f003:**
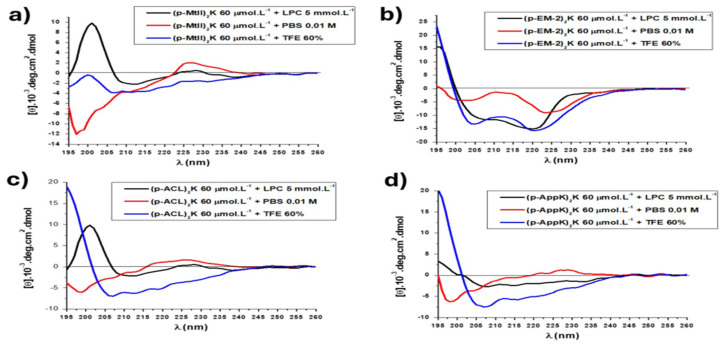
CD spectra of the (p-MtII)_2_K (**a**), (p-EM-2)_2_K (**b**), (p-ACL)_2_K (**c**), and (AppK)_2_K (**d**) peptides in aqueous solution (PBS), in structuring solution (TFE 60%) and in solution containing micelles (LPC).

**Figure 4 toxins-16-00308-f004:**
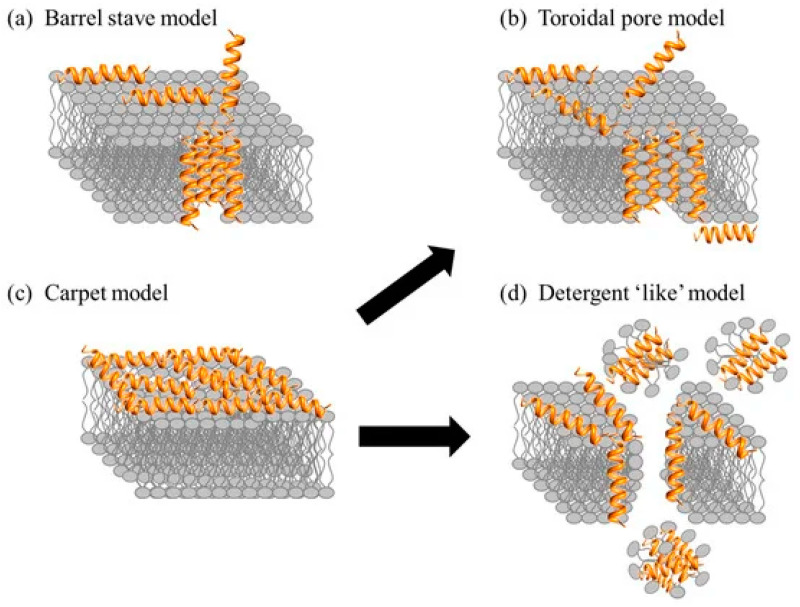
Mechanisms of action of AMPs: barrel-stave (**a**), toroidal pore (**b**), carpet-like (**c**), detergent-like (**d**). Reproduced from Kumar et al. [30], published by *Biomolecules*, MDPI, 2018, under the CC BY license. http://creativecommons.org/licenses/by/4.0/ (accessed on 3 July 2024).

**Figure 5 toxins-16-00308-f005:**
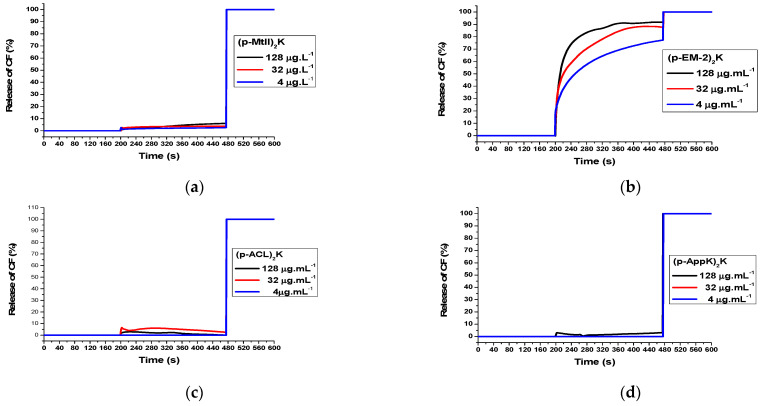
Carboxyflourescein release promoted by peptides from large unilamellar vesicles (LUVs) containing 15 mM POPC at different concentrations of peptides.

**Figure 6 toxins-16-00308-f006:**
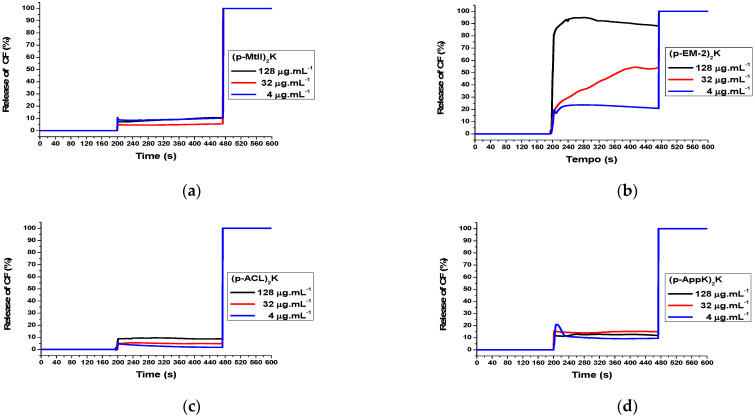
Permeabilization of 12 mM POPC (80%) + 3 mM POPG (20%) vesicles at different concentrations of peptides.

**Figure 7 toxins-16-00308-f007:**
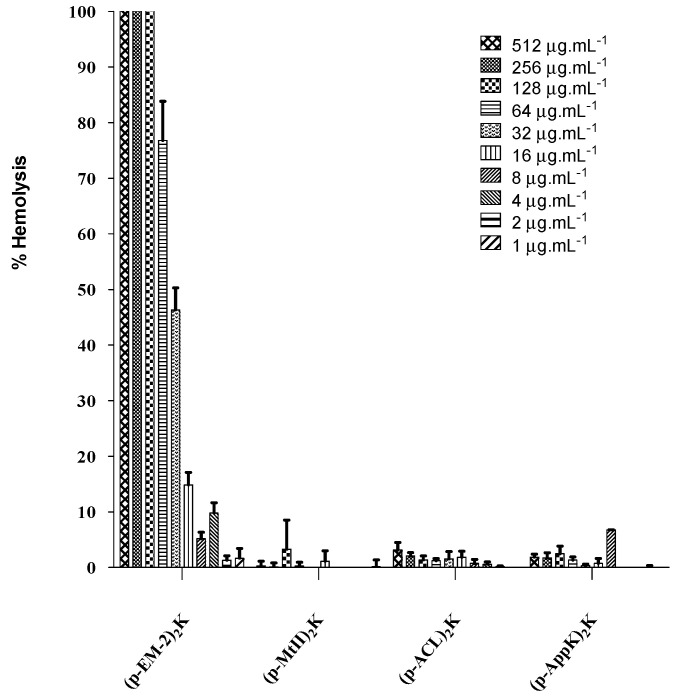
Hemolytic activity of the peptides (p-MtII)_2_K, (p-EM-2)_2_K, (ACL)_2_K, and (AppK)_2_K at different concentrations of peptides.

**Table 1 toxins-16-00308-t001:** Minimum inhibitory concentration (MIC) and minimum bactericidal concentration (MBC) for the peptides (p-MtII)_2_K, (p-EM-2)_2_K, (p-ACL)_2_K, (p-AppK)_2_K, and (p-BthTX-I)_2_K (Santos-Filho et al. (2021a) [1]).

Bacterial strains	(p-MtII)_2_K		(p-EM)_2_K		(p-ACL)_2_K		(p-AppK)_2_K		(p-BthTX-I)_2_K * Santos-Filho et al., 2021 [1]	
	MIC (μg·mL^−1^)	MBC (μg·mL^−1^)	MIC (μg·mL^−1^)	MBC (μg·mL^−1^)	MIC (μg·mL^−1^)	MBC (μg·mL^−1^)	MIC (μg·mL^−1^)	MBC (μg·mL^−1^)	MIC (μg·mL^−1^)	MBC (μg·mL^−1^)
*S. epidermidis* ATCC 35984	16	32	16	16	8	8	8	4	16	64
*S. aureus* ATCC 25923	256	256^¨^	32	32	32	32	32	64	128	256
*S. aureus* ATCC 8095	256	256	32	32	64	64	64	256	N.D.	N.D.
*E. faecalis* ATCC 29212	128	>512	32	32	32	128	64	128	128	>512
*E. faecium* ATCC 700221	16	>512	16	16	16	64	16	128	32	>128
*K. pneumoniae* ATCC 700603	256	>512	512	256	16	64	16	16	256	256
*E. coli* ATCC 25922	128	512	32	32	16	32	16	128	64	128
*A. bauamannii* ATCC 19606	256	>512	32	32	32	16	16	32	256	>512
*P. aeruginosa* ATCC 27853	N.D.	N.D.	N.D.	N.D.	64	128	64	512	>512	N.D.

N.D.: not determined; This test was not carried out because the compound did not show activity against the strain. The table shows Gram-positive bacteria strains in white and Gram-negative strains in gray * Adapted from Santos-Filho et al., 2021 [1].

## Data Availability

Data supporting the conclusions of this study can be made available upon request from the corresponding author.

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
