# Peer review of "Synthesis, Characterization, and Study of the Antimicrobial Potential of Dimeric Peptides Derived from the C-Terminal Region of Lys49 Phospholipase A2 Homologs"

_toxins, 2024, doi:10.3390/toxins16070308_

Round 1

Reviewer 1 Report

Comments and Suggestions for Authors

1. General comment

In the manuscript, the author investigated the antimicrobial activity against ESKAPE strains, hemolytic activity, second structure and permeabilization of lipid vesicles of four peptides (p-MtII)2K, (p-EM-2)2K, (p-ACL)2K and (p-AppK)2K and found that (p-ACL)2K and (p-AppK)2K showed high antimicrobial activity and low hemolytic activity. Future studies are expected to evaluate the possible activity against multidrug-resistant bacterial strains, yeasts, protozoa or other microorganisms.  

2. Major revision

1) Figures 3, 4 and 5

It is strongly recommended to enlarge the figures and use thicker fonts and lines.

2) Lines 264~279

It is recommended to add the figures of "barrel-stave", "toroidal pore", "carpet-like" and “detergent-like” models, so that the reader of Toxins can easily understand the interaction of AMPs with lipid membranes.

3. Minor revision

1) Line 178: Revise “(CBM)” to “(MBC)”.

2) Line 264: Revise “PAMs” to “AMPs”.

Author Response

Dear Editor,

We are sending the revised version of the manuscript toxins-3048748 entitled " Synthesis, characterization, and study of the antimicrobial po-tential of dimeric peptides derived from the C-terminal region of Lys49 phospholipase A2 homologs".

All of the changes suggested by the reviewers have been accepted and incorporated in the text. The responses to the reviewers’ questions and comments are also attached. We believe that the corrections inserted to the manuscript have improved it considerably, and we hope that you will now find it suitable for publication in Toxins.

Sincerely,

Reply to Reviewers

Reviewer #1: In the manuscript, the author investigated the antimicrobial activity against ESKAPE strains, hemolytic activity, second structure and permeabilization of lipid vesicles of four peptides (p-MtII)2K, (p-EM-2)2K, (p-ACL)2K and (p-AppK)2K and found that (p-ACL)2K and (p-AppK)2K showed high antimicrobial activity and low hemolytic activity. Future studies are expected to evaluate the possible activity against multidrug-resistant bacterial strains, yeasts, protozoa or other microorganisms.

  1. Major revision

1) Figures 3, 4 and 5

It is strongly recommended to enlarge the figures and use thicker fonts and lines.

A.: Thank you the reviewer for pointing this out. Figures were changed for better visualization

2) Lines 264~279

It is recommended to add the figures of "barrel-stave", "toroidal pore", "carpet-like" and “detergent-like” models, so that the reader of Toxins can easily understand the interaction of AMPs with lipid membranes.

A.: Thank you the reviewer for pointing this out. Figure 2 was added as requested.

  1. Minor revision

1) Line 178: Revise “(CBM)” to “(MBC)”.

A.: Done

2) Line 264: Revise “PAMs” to “AMPs”.

A.: Done

Reviewer 2 Report

Comments and Suggestions for Authors

The work is interesting and concerns the important problem of searching for new effective molecules with antimicrobial activity against the drug-resistant strains that exhibit low cytotoxicity and hemolysis. Authors synthesized and characterized the dimeric peptides derived from the C-terminal region of Lys49 phospholipase A2 homologs, which appear to be promising antibacterial biomolecules with low hemolytic activity. A minor suggestion concerns the spelling of Gram-positive and Gram-negative bacteria should be written in capital letters, please correct it.

Author Response

Dear Editor,

We are sending the revised version of the manuscript toxins-3048748 entitled " Synthesis, characterization, and study of the antimicrobial po-tential of dimeric peptides derived from the C-terminal region of Lys49 phospholipase A2 homologs".

All of the changes suggested by the reviewers have been accepted and incorporated in the text. The responses to the reviewers’ questions and comments are also attached. We believe that the corrections inserted to the manuscript have improved it considerably, and we hope that you will now find it suitable for publication in Toxins.

Sincerely,

Reply to Reviewers

Reviewer #2: The work is interesting and concerns the important problem of searching for new effective molecules with antimicrobial activity against the drug-resistant strains that exhibit low cytotoxicity and hemolysis. Authors synthesized and characterized the dimeric peptides derived from the C-terminal region of Lys49 phospholipase A2 homologs, which appear to be promising antibacterial biomolecules with low hemolytic activity. A minor suggestion concerns the spelling of Gram-positive and Gram-negative bacteria should be written in capital letters, please correct it.

A.: We would like to thanks referee for this comment. The correction requested was performed

Reviewer 3 Report

Comments and Suggestions for Authors

see attachment

Comments on the Quality of English Language

see above

Author Response

Dear Editor,

We are sending the revised version of the manuscript toxins-3048748 entitled " Synthesis, characterization, and study of the antimicrobial po-tential of dimeric peptides derived from the C-terminal region of Lys49 phospholipase A2 homologs".

All of the changes suggested by the reviewers have been accepted and incorporated in the text. The responses to the reviewers’ questions and comments are also attached. We believe that the corrections inserted to the manuscript have improved it considerably, and we hope that you will now find it suitable for publication in Toxins.

Sincerely,

Reply to Reviewers

Reviewer #3: The manuscript “Synthesis, characterization, and study of the antimicrobial potential of dimeric peptides derived from the C-terminal region of Lys49 phospholipase A2 homologs” deals with the antimicrobial activity of 4 peptides obtained through modification of molecules derived from the C-terminal region of PLA2-like proteins from snake venom. The proposed modifications consist in a dimerization methodology previously applied by the authors to the p-BthTX-I peptide from bothropstoxin I, that gave origin to the molecule called (p-BthTX-I)2K.

The manuscript fits in the consolidated line of research aimed at identifying peptides with antimicrobial activity, in particular against antibiotic-resistant bacteria recognized as a threat to human health.

Although the novelty of the contribution is scarce, being the main goal of the manuscript the evaluation of the reproducibility of the previously proposed procedure of modification to other similar peptides, the addition of new molecules to the potential candidates for the development of novel antimicrobials is always interesting. The obtained molecules indeed show some antimicrobial activity, but it is curious that this activity is not compared to the one of the relative monomeric molecules, on the contrary it is stated that “Future assays comparing the activity of the dimers obtained through this strategy with the monomers of these peptides should be carried out” (lines 23-25, Abstract). Furthermore, the presented results should be considered preliminary to a deeper investigation on the mechanism of action of these molecules.

A general concern is related to the structure of the manuscript. Although the presentation of Results and Discussion in a single section (if permitted by the journal) can be accepted and sometimes useful, in this case it gives raise to considerable redundancy. In my opinion, this section should be simplified, avoiding repetitions and focusing on the relevant points (see below specific comments).

Specific comments

Introduction

Lines 36-38: “AMPs offer a broad spectrum of action, high specificity, and low propensity of selecting microbial resistance [2,4,5].” While ref.2 fits with the sentence, refs 4 and 5 could (should) be substituted by relevant general reviews on AMPs selected among the many papers recently published on the topic.

A.: Thank you the reviewer for pointing this out. Refs 4 and 5 were replaced for:

  • Hancock, R.E.. Cationic peptides: effectors in innate immunity and novel antimicrobials. Lancet Infect Dis. 2001, 1, 156-164.
  • Li, X.; Zuo, S.; Wang, B.; Zhang, K.; Wang, Y. Antimicrobial Mechanisms and Clinical Application Prospects of Antimicrobial Peptides. Molecules. 2022, 27, p.2675.

Results and discussion

A first general comment refers to cited references. Since, in the order of presentation, section 2. Results and Discussion and section 3. Conclusions come before section 4. Materials and methods, the cited references should be numbered and appear in the text and in References section in the same correct order. Currently, in the text there is a gap between references 17 (the last cited in Introduction) and reference 21 (the first cited in in Results and Discussion section) while references 18, 19, and 20 appear in the text in Materials and methods at the end of the manuscript.

A.: We would like to thank referee for this comment. The article was revised and order of references was corrected.

A second general comment also refers to the order of presentation of manuscript sections. The anticipation in section 2 of some methodological details and the explanation of the acronyms (now presented in section 4. Materials and methods), would help the readership by allowing a fluent reading and a better understanding.

A.: Thank you the reviewer for pointing this out. Text was changed for better understanding

Line 83: Agkistrodon piscivorus piscivorus piscivorus

A.: Corrected

Line 84: myotoxins instead of mycotoxins

A.: Done

Line 95: the reference number should be added after (Mendes et al. 2019)

A.: Done

Line 103 (Figure 1 Legend): Renetseder et al. (1985 [24]) and Line 121: Bitencourt et al. (2023) [25]. While in the text the reference numbers 24 and 25 are correct, in the References section the cited papers are reversed

A.: We would like to thank referee for this comment. The article was revised and order of references was corrected.

Line 111: the reference number should be added after Santos-Filho and collaborators (2021a)

A.: Done

Line 121: observed that the dimerization

A.: Done

Line 123: …significant reduction in, bacterial activities this activity.

Line 156: instead of “…and these deaths will become the leading cause of death worldwide” it would be better “…making infectious diseases the leading cause of death worldwide”

Lines 175-176: “…antimicrobial activity tests were performed on the same strains to compare the performance of the dimers. The results of this activity are shown in Table 1.” Since it is understood that were used the same strains as in the previous papers for comparative purposes, it would be desirable a direct comparison of the obtained results. Therefore, the suggestion is to add a column in Table 1 showing the results previously obtained with (p-BthTX-I)2K (obviously mentioning the relative references). This addition could allow to delete the long following paragraph (lines 184-195), gaining clarity and ease of reading.

Line 178 (Table 1 title): MBC instead of CMB

Lines 180-182 (Table footnote): “*Represents bactericidal activity, and # represents bacteriostatic activity. The activity is defined by the MBC/MIC ratio. For results = 4 or < 4 the activity is considered bactericidal, according to Pankey 181 and Sabath (2004).” Since the reported definition is questionable and not present in standard documents, it is suggested to delete this sentence and to leave only the values in the table.

Line 208: CFUs instead of UFC

Line 229: the reference number should be added after Almeida and co-workers (2022)

Lines 245-246: “Although numerous studies have highlighted the potential of these biomolecules, the correlation between their structure and activity remains poorly understood [13].” Since the contribution of this sentence in the text is not essential, deletion is suggested.

Lines 249 and 253: It would be better to indicate the meaning of the acronyms LPC and TFE (since they appear here for the first time)

Lines 264-266: The following modification of the text is suggested “The mechanism of action of PAMs is still unclear. Different models, such as the “barrel-stave” and “toroidal pore” models, have been proposed to describe the interaction of AMPs with lipid membranes.”

Line 278: “…the mechanism of action of many AMPs remain enigmatic.”

Lines 281-288: the duplicated sentences must be deleted (leave lines 289-296)

Lines 298 and 299: It would be better to indicate the meaning of the acronyms POPC and POPG (since they appear here for the first time)

Line 307: LUV acronym is not defined and appears only in the legend of Figure 4 Note also that in the graphics of Figures 4 and 5 x and y axis legends are tempo and intensidade %

Lines 319-321: “The release of CF from bacterial membrane mimetics demonstrated leakage of approximately 10 to 20% for the peptides (p-MtII)2K (Figure 5a), (p-ACL)2K (Figure 5c), and (p-AppK)2K (Figure 5d),…” why this sentence is anticipated, since the results on bacterial membrane are described later (starting from line 327)? This anticipation is confusing, a change is suggested

Lines 322-326: “The interaction profile combined with the structural conformations adopted by the molecules in membrane mimetics (LPCs) and the fact that these molecules present antimicrobial activity, even if they do not rupture or form permanent pores in the vesicles, may suggest the possible formation of transient pores. Further studies must be carried out to confirm this hypothesis.” Formation of transient pores can allow entry of the peptides in the intracellular environment. A deeper investigation is surely needed to clarify the possible intracellular targets and related mechanisms of action of these peptides (as reported for previously described AMPs); a brief comment on this point could be added

Lines 345-346 and lines 359-361 repeat the same statement. One of the sentences should be deleted

Materials and Methods

Lines 409-413: Although the bacteria are reference strains, the indication of their susceptibility (or resistance) to conventional antibiotics would be useful for the reader. Moreover, no indication of the amount of bacterial inoculum is given

Lines 427-428: “MBC/MIC values less than or equal to 4 and greater than 4 indicate bactericidal and bacteriostatic activities, respectively” in agreement with the previous comment (lines 180-182), deletion of this sentence is suggested.

- The "Highlights" Section does not reflect the totality of the study and should be rewritten including the findings.

A.: Highlights were reformulated

- Line 84 - Correct as: … range of bacterial strains, including …

A.: Done

- Line 88 - Correct as: … are a new group that have …

A.: Done

- Line 114 - MIC plates were analyzed visually. However, as showed in Fig. 1, the readings were recorded as ODs (absorbance 600 nm), as recommended by standard methodology. Please clarify.

A.: Thank you the reviewer for pointing this out. We did both: analyzed the plates visually and by OD. We have corrected the statement in the text to clarify the information.

- Line 120 - It should be indicated to the reader which peptide was carboxyfluorescein-labeled, and how this labeling was performed.

A.: The carboxyfluorescein-labeled peptide was indicated for a better understanding. The labeling protocol was briefly described on “results and discussion” section, as follow: CF was attached to the amino-terminal portion of the peptide using Fmoc-SPPS methodology using the same protocol established for the other amino acids. The peptide containing CF was cleaved from the resin, dimerization was carried out by cysteine oxidation, and was then purified to obtain the pure peptide CF-(p-BthTX-I)2.

- Line 143 - Reference(s) to support the assay with Annexin-V should be added.

A.: Done

- Line 144 - Describe which peptide was used.

A.: Done

- Line 202 - The word 'since' is more appropriate than 'once'.

A.: Done

- Line 209 - Use 'antimicrobial peptides' instead of AMPs, since this acronym does not appear throughout the text.

A.: Done

- Line 255 - Correct 'coupled' instead of 'coupling'.

A.: Done

- Line 266 - The sentence is confusing. What do the authors mean by 'although of the cell size and granularity ?

A.: This statement was improved for a better understanding

- Line 302 - What is the meaning of "comportment"

A.: This statement was improved for a better understanding. It is important to cite that the article was forward to professional editors at Editage, a division of Cactus Communications, to ensure that the English language is clear and free of errors.

- Line 316 - The reference Bayles, 2014 is missing in the "References" Section.

A.: The reference was added in the “reference” section.

- Line 326 - Pec et al. (2003) is a reference for apoptosis in eukaryotic cells.

A.: This statement was improved for a better understanding. Actually, Pec et al. (2003) were cited in order to exemplify late apoptosis. 

- Line 334 - Correct as: … of viable E. coli after treatment found …

A.: Done

- Line 334 - Change 'small velocity' by 'delayed action'.

A.: Done

- Line 370 - Correct as: After 5 min …

A.: Done

- Line 371 - Correct as: … (Figure 7G-H and 8G-H) it is already possible to see

A.: Done

- Line 426 - Correct: E. coli and S. aureus.

A.: Done

- Figure 3 - Correct the y-axys if it corresponds to Fluorescence Intensity Units.

A.: Corrected

- Figure 4 - Standardize Escherichia coli or E. coli / Staphylococcus aureus or S. aureus here and in all Figures.

A.: Done

- Figure 6 - The S. aureus graph: correct "Necrosis" in the x-axys.

A.: Corrected

Reviewer #2:

Reviewer #2: In the present submitted manuscript, Santos-Filho et al. take a rather extensive approach to elucidating the mechanism of action for the pBthTX-I against E. coli and S. aureus. The methodology appears appropriate and the results are interesting. It is recommended that the authors take some additional time for English language editing, as there are numerous sentences that are poorly written and cause confusion for the reader. Below are detailed the issues for consideration.

  1. Sentences that need rewriting for clarity:

    Lines 48-50; 94-96; 224-227; 264-267; 318-320; 434-436

A.: All these sentences were improved for a better understanding. It is important to cite that the article was forward to professional editors at Editage, a division of Cactus Communications, to ensure that the English language is clear and free of errors.

  1. Gram is a proper name and should be capitalized in all instances throughout the manuscript.

A.: Done

  1. Line 69: PLA2-homologue instead of homologous.

A.: Done

  1. Line 74: a word appears to be missing between "main...??...responsible".

A.: This statement was improved for a better understanding.

  1. Line 88: remove "for"

A.: Done

  1. Line 298: add "be" between "could quenching"

A.: Done

  1. Lines 348-359: I suggest revising the terminology used to refer to the layers/structures external to the bacterial plasma membrane. The cell wall in bacteria refers only to the layer of peptidoglycan and the macromolecules therein. The cell envelope is more appropriately used when referring to all external layers. In this case, the authors would be better suited to use cell envelope rather than cell wall to avoid confusion and use the most applicable terminology.

A.: Thank you the reviewer for pointing this out. The term cell wall was replaced to cell envelope

  1. Line 357: LTA instead of LPA?

A.: Done

  1. Line 400: disrupt and not disrupter

A.: Done

  1. Line 406: established instead of establishing

A.: Done

  1. Line 488: Annexin and not Apexin

A.: Done

  1. Lines 491 and 495: correct figure legend such that (D, F, G) becomes (D, F, H) to correctly refer to the image.

A.: Done

  1. Line 498: Correct the unit of concentration for the LTA.

A.: Done

Round 2

Reviewer 3 Report

Comments and Suggestions for Authors

The authors addresses the issues raised by the reviewer.

The manuscript has been sufficiently improved.

Comments on the Quality of English Language

see above